# Evaluating extraction methods to study canine urine microbiota

**Ryan Mrofchak[1], Christopher Madden[1], Morgan V. Evans[1,2], Vanessa L. Hale**[1] *

**1** Department of Veterinary Preventive Medicine, Ohio State University College of Veterinary Medicine, Columbus, Ohio, United States of America, **2** Division of Environmental Health Sciences, Ohio State University College of Public Health, Columbus, Ohio, United States of America

* hale.502@osu.edu

**Data Availability Statement:** Sequencing data is available at NCBI Bioproject PRJNA689589.

**Funding:** Support for this project was provided by the Ohio State University College of Veterinary Medicine Canine Funds, the Infectious Disease

## Abstract

The urinary microbiota is the collection of microbes present in urine that may play a role in host health. Studies of urine microbiota have traditionally relied upon culturing methods aimed at identifying pathogens. However, recent culture-free sequencing studies of the urine microbiota have determined that a diverse array of microbes is present in health and disease. To study these microbes and their potential role in diseases like bladder cancer or interstitial cystitis, consistent extraction and detection of bacterial DNA from urine is critical. However, urine is a low biomass substrate, requiring sensitive methods to capture DNA and making the risk of contamination high. To address this challenge, we collected urine samples from ten healthy dogs and extracted DNA from each sample using five different commercially available extraction methods. Extraction methods were compared based on total and bacterial DNA concentrations and bacterial community composition and diversity assessed through 16S rRNA gene sequencing. Significant differences in the urinary microbiota were observed by dog and sex but not extraction method. The Bacteremia Kit yielded the highest total DNA concentrations (Kruskal-Wallis, $p$ = 0.165, not significant) and the highest bacterial DNA concentrations (Kruskal-Wallis, $p$ = 0.044). Bacteremia also extracted bacterial DNA from the greatest number of samples. Taken together, these results suggest that the Bacteremia kit is an effective option for studying the urine microbiota. This work lays the foundation to study the urine microbiome in a wide range of urogenital diseases in dogs and other species.

## Introduction

Urine, in the absence of urinary tract infection, has long been considered sterile; a principle still taught in many healthcare professional settings to date. However, evidence counter to this idea has been accumulating for several decades [1]. Culture-positive asymptomatic bacteriuria is commonly reported in women and older adults; although, this is sometimes deemed "contamination" based on bacterial counts $< 10^5$ [2]. Culture-negative symptomatic urinary tract infections (UTIs) are also common, and can, in some cases, be attributed to fastidious organisms that fail to grow using standard urine culturing (SUC) techniques [2–8]. More recently, culture-independent next-generation sequencing of urine and enhanced quantitative urine

Institute, and the Department of Veterinary Preventive Medicine (RM, CM, MVE, VLH). MVE was additionally supported by Ohio State University College of Public Health Collaborative Postdoctoral Research funding. The funders had no role in study design, data collection and analysis, decision to publish, or preparation of the manuscript.

**Competing interests:** The authors have declared that no competing interests exist.

culture (EQUC) has revealed the presence of bacteria in >90% of individuals–including those with and without UTIs and from urine collected via free-catch, transurethral catheter, or suprapubic aspirates [5–7, 9–14]. Collectively, this work provides evidence for the presence of a urinary microbiota containing live bacteria that are present in healthy individuals.

A growing body of work has revealed profound links between the microbiota (oral, gut, lung, vaginal) and host health [15–19]–from nutrient metabolism [20], to immune development and defense [21], and colonization resistance [22]. Thus, it follows that the urine / bladder microbiome may also have a critical role in host health, but it has been far less studied than the microbiota of the aforementioned body regions. Notably, the "urine is sterile" dogma contributed to the exclusion of urine in the first phase of Human Microbiome Project (HMP) launched in 2007 [23]. In 2014, the second phase of the HMP launched and included the urine / bladder microbiome. However, work on the urine microbiome continues to lag. Although urine is not sterile, it has a low microbial biomass, making it more challenging to characterize and at greater risk for contamination [24]. Despite this, several recent studies on urine have identified clear shifts in the microbial community associated with age [10, 25–30], sex [12, 31, 32], urgency urinary incontinence in women [10, 33–35], bladder cancer [31, 36–42], interstitial cystitis [43–49], neuropathic bladders [50, 51] and pneumonia [52]. However, additional work is needed to identify definitive and mechanistic links between the urine / bladder microbiome, host health, and urinary tract disease.

Urinary tract disease is one of the most common diagnoses in veterinary medicine [53, 54]. In addition, dogs are a valuable translational model for many human diseases, including urogenital diseases like bladder cancer [55]. However, there have only been two studies, to our knowledge, characterizing canine urine microbiota [56, 57] and none evaluating canine urine DNA extraction methods. Multiple methods of urine microbial DNA isolation have been reported in human studies [12, 51, 58–61], and there are a few studies that have compared microbial DNA extraction methods. These studies include a comparison of methods for extracting fungal [62] and viral DNA from urine [59, 63, 64] as well as a study on methods to reverse crystal precipitates that interfere with DNA extraction [65]. Yet, only one recent study has evaluated urine DNA extraction methods in relation to the bacterial microbiota in humans [61], and there are no comparable studies in dogs. In this study, our objective was to compare canine urine bacterial DNA quantity and 16S rRNA sequencing results of five commonly used extraction methods. These extraction methods included four DNA isolation kits from Qiagen —Bacteremia, Blood and Tissue, PowerFecal®, and PowerFecal® Pro—and an extraction protocol using magnetic beads.

## Materials and methods

### Sample collection

This study was approved by The Ohio State University College Institutional Animal Care and Use Committee (IACUC #2019A00000005). Mid-stream free catch urine samples were collected from 10 healthy dogs in September 2019 at The Ohio State University College of Veterinary Medicine (Columbus, OH, USA) with owner consent. Enrollment criteria included: a body weight of at least 30 lbs (~13.6 kg) and the ability to produce > 30 ml of urine in a single urination. Dogs were excluded if they had any history of urinary tract disease or antibiotic use within three weeks of sample collection. We collected urine samples from a total of 10 dogs, including 4 males (3 neutered) and 6 females (5 spayed). The average age of the dogs was 3.7 years old (range: 0.75–10 years) and represented multiple breeds including: one Great Pyrenees, one Labrador Retriever, one Golden Retriever, and seven mixed breed dogs. Full metadata on each dog is available in **S1 Table**. All urine samples were stored on ice immediately

after collection and transported to the laboratory within 6 hours of urination. Samples were then transferred into a -80˚C freezer where they remained until extraction.

## DNA extraction

Five extraction methods were tested: QIAamp® BiOstic® Bacteremia DNA Kit (B), DNeasy® Blood and Tissue Kit (BTL), QIAamp® PowerFecal® DNA Kit (PF), QIAamp® PowerFecal® Pro DNA Kit (PFP) (Qiagen, Venlo, Netherlands) and an extraction protocol using magnetic beads (MB) [66]. Each protocol incorporated varying chemical, mechanical, and thermal lysing steps to facilitate DNA extraction (Table 1).

To prepare urine samples for extraction, 3.0 ml of urine was centrifuged at 4˚C and 20,000 x $g$ for 30 minutes. Samples that underwent MB extraction were centrifuged at 4˚C and 20,000 x $g$ for 20 minutes. After centrifuging, the supernatant was removed and discarded and the pellet was used for extraction. Samples were assigned initials unique to each dog (AW, CB, CS, DD, DH, HB, LS, SF, SM, and ZR). Extraction methods were abbreviated as seen in Table 1 (B, BTL, MB, PF, or PFP). Negative and positive controls were also extracted from each method. The negative control was a blank (no sample) tube that underwent the full extraction process. The positive control was 3.0 ml of urine from dog AW spiked with 3 x $10^8$ CFUs of *Melissococcus plutonius*. *M. plutonius* is a honeybee pathogen that would not be expected in the urine / gut of dogs [67]. Brief descriptions of each extraction method are included below.

**Bacteremia (B).** Urine pellets were resuspended in a lysis buffer (MBL) and placed in a 70˚C water bath for 15 minutes. Samples then underwent two rounds of bead beating (6 m/s for 60 seconds with a 5 minute rest between rounds). Bead beating was performed on a MP FastPrep-24™ 5G (MP Biomedicals, Santa Ana, California, USA). After bead beating, the samples were cleaned with an Inhibitor Removal Solution. The remainder of the protocol was followed with two modifications. First, centrifugation was performed at 13,000 x $g$ instead of 10,000 x $g$. Second, during the final step, DNA was eluted into 50 µl of elution buffer (EB) and incubated at room temperature for five minutes; then, the first eluate was run through the silica membrane of the spin column a second time to maximize DNA yield.

**Blood and Tissue with Lysozyme (BTL).** Urine pellets were resuspended in a lysis buffer adapted from Pearce *et al.*, 2014. The lysis buffer consisted of 10mM tris, 1mM EDTA, 1.0% SDS, pH 8.0, and 20 mg/mL lysozyme (Sigma Aldrich, St. Louis, MO) [10, 29, 68, 69]. The

**Table 1. Mechanical, chemical and thermal lysis properties of each extraction method.**

| Kit | Mechanical lysis? | Mechanical lysis method | Chemical lysis? | Chemical lysis method | Thermal lysis? | Thermal lysis method |
|---|---|---|---|---|---|---|
| Bacteremia (B) | yes | bead beating | yes | Lysis buffer (MBL) | yes | 70˚C heat block / water bath for 15 min. |
| Blood and Tissue with Lysozyme (BTL) | no | N/A | yes | Lysozyme (10mM EDTA, 1.0% SDS, 20 mg/ml lysozyme) | yes | 37˚C heat block / water bath for 1 hour (with lysozyme) |
| Magnetic Beads (MB) | no | N/A | yes | 10mM tris, 2mM EDTA, and 1% SDS | yes | 65˚C water bath for 30 min. Three freeze / thaw cycles–liquid nitrogen for 1 minute followed by 65˚C for 5 minutes. |
| PowerFecal® (PF) | yes | bead beating | yes | Lysis buffer (Powerbead Solution) | yes[a] | N/A |
| PowerFecal® Pro (PFP) | yes | bead beating | yes | Lysis buffer (CD1) | no | N/A |

Mechanical, chemical and thermal lysis properties of each extraction method.

[a]This manufacturer protocol includes a thermal lysing step; however, we used a modified version of this protocol, skipped the thermal lysing step, and performed two rounds mechanical lysing using a bead beater rather than the vortex adapter indicated in the protocol.

urine pellet and lysis buffer were then incubated in a 37 °C water bath for 1 hour. The remainder of the extraction protocol was followed per manufacturer instructions with one modification. In the final step, DNA was eluted in 50 µl of elution buffer (AE), incubated at room temperature for five minutes; then, the first eluate was run through the silica membrane of the spin column a second time.

**Magnetic Beads (MB).** Per Liu *et al.* (2017), urine pellets were resuspended in a lysis buffer composed of 10mM tris, 2mM EDTA, and 1% SDS, pH 8.0. The suspension was then frozen in liquid nitrogen for 1 minute followed by incubation in a 65°C water bath for 5 minutes; the freeze / thaw process was repeated three times. After the third freeze / thaw step, suspensions were incubated for 30 minutes in a 65°C water bath. Suspensions were then centrifuged at 20,000 x *g* for five minutes. After completing the lysis step and centrifugation, the supernatant was placed in PCR tube strips containing AMPure XP magnetic beads (Beckman Coulter, Indianapolis, IN). The supernatant and magnetic beads were homogenized and incubated at room temperature, then placed on a magnetic separator for 5 minutes. During this step, lysed DNA was drawn to magnetic beads in the tube strips. The remaining supernatant was removed and beads were washed with 80% ethanol. This was repeated twice. After washing, DNA-bound beads were dried in a 37°C heat block for 15 minutes [66]. Dried DNA-bound beads were then resuspended in 40 µl Qiagen© C6 solution. Resuspended samples were then placed on a magnetic separator for 1–2 minutes to pellet beads. The resulting supernatant contained DNA used in downstream analyses.

**PowerFecal®(PF).** Urine pellets were resuspended in lysis buffer (PowerBead Solution + C1) and subjected to two rounds of bead beating (6 m/s for 60 seconds with a five-minute rest between cycles). Samples then underwent multiple inhibitor removal and purification steps. The remainder of the extraction protocol was followed with two modifications. During the second round of centrifuging, after applying the ethanol-based wash solution (C5) to the spin column, samples were centrifuged for 2 minutes (instead of 1 minute) to remove residual wash solution. In the final step, DNA was eluted in 50 µl of elution buffer (C6), incubated at room temperature for five minutes; then, the first eluate was run through the silica membrane of the spin column a second time.

**PowerFecal® Pro (PFP).** Urine pellets were resuspended in lysis buffer (CD1) and subjected to two rounds of bead beating (6 m/s for 60 seconds with a five-minute rest between cycles). The remainder of the extraction protocol involving multiple inhibitor removal and purification steps was followed as written with one modification. In the final step, DNA was eluted in 50 µl of elution buffer (C6), incubated at room temperature for five minutes; then, the eluent was run through the silica membrane of the spin column a second time.

DNA yields from all samples were then quantified on a Qubit® 4.0 Fluorometer (Invitrogen, Thermo Fisher Scientific™, Carlsbad, CA, USA) using a 1X dsDNA High Sensitivity Assay. Hereafter, DNA concentrations measured using Qubit® are referred to as total DNA concentrations.

## Quantification of bacterial DNA by qPCR

Bacterial DNA was amplified using 16S rRNA bacterial primers and probes per Nadkarni *et al.* (2002) on a QuantStudio™ 3 Real-Time PCR System (Applied Biosystems™, Thermo Fisher Scientific™, Carlsbad, CA, USA). 300 nM of forward primer (5′–TCCTACGGGAGGCAGC AGT– 3′), 300 nM of reverse primer (5′–GGACTACCAGGGTATCTAATCCTGTT– 3′), and 175 nM of probe ((6FAM)– 5′–CGTATTACCGCGGCTGCTGGCAC– 3′–(TAMRA)) were added to each reaction. qPCR cycling parameters were as follows: 50°C for 2 min, 95°C for 10 min (initial denaturation) and 40 cycles of 95°C for 15 s (denaturation) and 60°C for 1 min

(annealing and extension) [70]. To be included in analysis, at least two replicates per sample had to amplify. Following qPCR, cycle thresholds were $\log_{10}$-transformed using the equation listed below under "qPCR standard curve," and the antilog of each sample was used to calculate the bacterial DNA concentration in each sample.

## qPCR standard curve

DNA extracted from an *Escherichia coli* isolate was used to generate a standard curve for qPCR. Five ten-fold dilutions of *E. coli* DNA ranging from approximately 5 x $10^4$ pg/µl to 5 x $10^{-1}$ pg/µl were run in triplicate using the primers, probe, cycling parameters, and the Quant-Studio instrument described above. DNA concentrations were then $\log_{10}$-transformed and plotted against cycle threshold values on a linear regression using R package ggplot2 v.3.3.2. The resulting equation was y = -5.329x + 36.504 ($R^2$ = 0.984) where y is the cycle threshold and x is the $\log_{10}$-transformed DNA concentration. To calculate estimates of the absolute cell counts in each sample, *Escherichia coli* was used as a standard, and the bacterial DNA concentrations were divided by the theoretical weight of one *E. coli* cell (4.96 fg DNA) per Nadkarni et al. 2002. Estimates of the absolute cell counts were then multiplied by relative abundances of specific taxa (e.g. *Sphingomonas* and *Pasteurellaceae* bacterium canine oral taxon 272) to obtain and compare absolute cell counts of these taxa.

## 16S rRNA sequencing and sequence processing

DNA underwent library preparation and sequencing at Argonne National Laboratory. Library preparation was performed as follows: the V4 region of the 16S rRNA gene was amplified using primers 515F and 806R [71, 72]. PCR reactions (25 µL) contained 9.5 µL of MO BIO PCR Water (Certified DNA-Free), 12.5 µL of QuantaBio's AccuStart II PCR ToughMix (2x concentration, 1x final), 1 µL Golay barcode tagged Forward Primer (5 µM concentration, 200 pM final), 1 µL Reverse Primer (5 µM concentration, 200 pM final), and 1 µL of template DNA. The following PCR conditions were applied: 94°C for 3 minutes to denature the DNA, with 35 cycles at 94°C for 45 s, 50°C for 60 s, and 72°C for 90 s; with a final extension of 10 min at 72°C. Amplicons were then quantified using PicoGreen (Invitrogen) and a plate reader (Infinite® 200 PRO, Tecan). Equimolar volumes of the amplicons were pooled into a single tube. This pool was then cleaned using AMPure XP Beads (Beckman Coulter), and quantified with a fluorometer (Qubit, Invitrogen). After quantification, the molarity of the pool was diluted to 2 nM, denatured, and then diluted to a final concentration of 6.75 pM with a 10% PhiX spike. Amplicons were sequenced on an Illumina MiSeq (Lemont, IL, USA) using V2 chemistry with 2 x 250 bp paired-end reads and customized sequencing primers and procedures [71]. Sequencing data is available at NCBI Bioproject PRJNA689589.

Raw, paired-end sequence reads were processed using QIIME2 v. 2020.2 [73]. The DADA2 plugin was used to truncate reads at 230 bp and trim 33 bp from the left side of both forward and reverse reads [74]. These parameters were used to ensure primers and barcodes were removed and to denoise paired end reads. Taxonomy was assigned on the resulting amplicon sequence variants (ASVs) in QIIME2 using the Silva 132 99% database from the 515F / 806R classifier [75, 76]. We opted not to rarefy at 300 reads as this drastically decreased diversity and eliminated rare taxa. Rarefaction can also increase Type 1 errors and variance where overdispersion can mask differential abundance between samples [77]. To avoid these issues and account for rare taxa, an unrarefied table was used as input in α- and β-diversity metrics.

## Statistical analyses

Total DNA concentrations (as measured by Qubit) and bacterial DNA concentrations (as calculated from qPCR) were tested for normality using the Shapiro Wilk Normality Test in R version 3.5.2. DNA concentrations and 16S rRNA read numbers were analyzed using the Kruskal-Wallis Rank Sum Test. Statistical significance was achieved if the p-value was less than 0.05.

Prior to 16S rRNA sequencing analysis, singletons (ASVs with only one read in the entire dataset) were excluded. We then used the R package decontam and applied both the frequency and prevalence (at a 0.5 threshold) methods to identify and remove putative contaminants from our dataset (R package, v.1.10.0) [78]. All diversity metrics were computed using the R package phyloseq with a p-value cutoff of 0.05 adjusted using the Benjamini & Hochberg False Discovery Rates [79]. To analyze bacterial diversity, three α-diversity metrics were used: Observed Features (equivalent to richness), Shannon, and Simpson. Kruskal-Wallis Rank Sum Tests were used to compare α-diversity results to categorical variables of interest (extraction method, sex, dog). Post-hoc pairwise comparisons were calculated using Pairwise Wilcoxon Rank Sum Tests. To compare bacterial composition between groups (extraction method, sex, dog), three β-diversity metrics were used: Bray Curtis, Unweighted UniFrac, and Weighted UniFrac. A permutational analysis of variance (PERMANOVA) based on Euclidean distance matrices and 1000 permutations was used to quantify these differences (adonis2 function, R package vegan v.2.5.6). Multilevel pairwise comparisons for dog and extraction method were calculated using pairwise PERMANOVAs with 1000 permutations [80]. An Analysis of Composition in Microbiomes (ANCOM) in QIIME2 was used to identify differentially abundant taxa between groups (extraction method, sex, dog).

## Results

### Urine total DNA concentrations

We extracted DNA from a total of 10 dogs using 5 different methods. A single positive and negative control were also extracted using each kit. Total DNA concentrations were measured on a Qubit fluorometer (S2 Table) and were not normally distributed (Shapiro-Wilk Normality Test, $p = < 0.0001$). Twenty-six samples (by kit = 9 PFP, 6 MB, 4 B, 4 PF, 3 BTL; by dog = 4 AW, 1 AWS, 1 CB, 4 CS, 5 DD, 3 DH, 4 HB, 1 LS, 1 SF, and 2 SM) and all 5 negative controls (one per method) had concentrations that were too low to read ($< 0.01$ ng/μl). Quantifiable DNA concentrations ranged from 0.02 ng/μl to 1.37 ng/μl for all other samples including positive controls. The number of samples with quantifiable DNA varied by extraction method (Fig 1A). BTL extracted quantifiable DNA from nine urine samples (including the positive control). B and PF extracted DNA from eight urine samples (including the positive controls). MB and PFP extracted DNA from six and two samples, respectively (Fig 1A). Notably, no DNA was recovered from the spiked positive control sample using PFP. Total DNA concentrations did not differ significantly by method (Fig 1B, Kruskal-Wallis, $p = 0.165$), but did differ significantly by sex (Fig 1C, Kruskal-Wallis, $p = 0.0007$) and by dog (Fig 1D, Kruskal-Wallis, $p = 0.001$). Males had significantly higher DNA concentrations; however, no pairwise comparisons of DNA concentrations were significant by dog (S3 Table). B produced the highest average total DNA concentrations, followed by PF, BTL, MB, and PFP.

### Urine bacterial DNA concentrations

Bacterial DNA concentrations were measured in triplicate via qPCR (S2 Table) and were not normally distributed (Shapiro-Wilk Normality Test, $p = < 0.0001$). Twenty-two samples (by

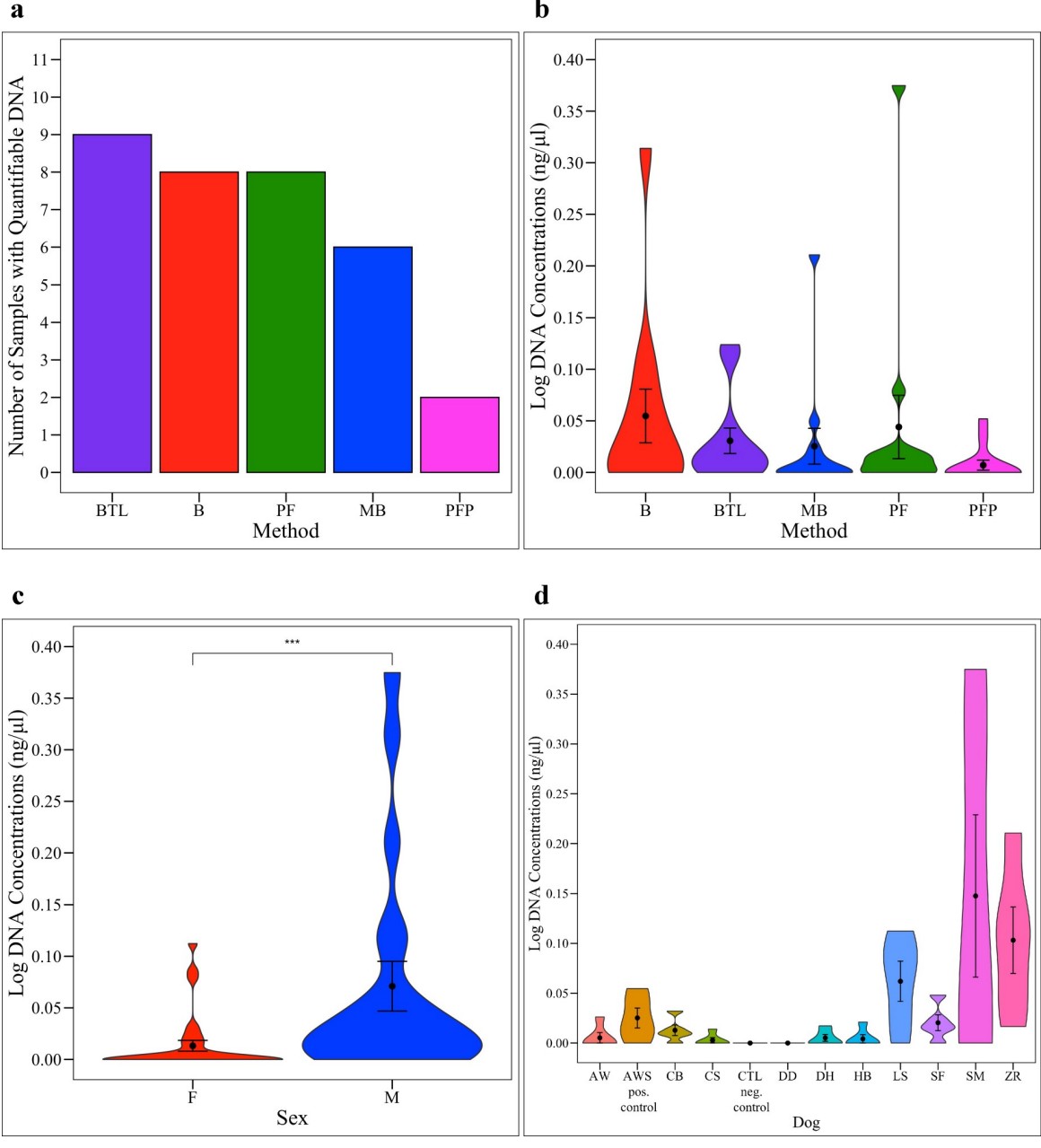

**Fig 1. Total DNA concentrations.** (**a**) Number of samples with measurable total DNA by extraction method (negative control excluded). Total DNA concentration, measured via Qubit, by (**b**) extraction method, (**c**) sex, and (**d**) dog. Total DNA concentrations did not differ significantly by extraction method (Kruskal-Wallis, $p = 0.165$) but did differ significantly by sex (Kruskal-Wallis, $p = 0.0007$; males > females), and dog (Kruskal-Wallis, $p = 0.001$). By dog, no pairwise comparisons were statistically significant. B = Bacteremia, BTL = Blood & Tissue with Lysozyme, MB = Magnetic Beads, PF = PowerFecal®, PFP = PowerFecal® Pro, F = Female, M = Male.

kit = 5 MB, 8 BTL, 5 PFP, 3 PF, 1 B; by dog = 2 AW, 1 AWS, 5 CB, 1 CS, 2 DD, 4 DH, 4 HB, 2 SF, and 1 SM) and all 5 negative controls did not amplify any bacterial DNA. In total, twenty-seven samples did not amplify bacterial DNA. Five samples (4 MB and 1 PFP) were excluded for failing to have at least two replicates amplify (**S2 Table**). All samples exhibited less than 3% variation in cycle threshold values between replicates with three exceptions: the spiked positive control extracted using MB (5.6% variation), and samples from dogs SF (6.1% variation) and

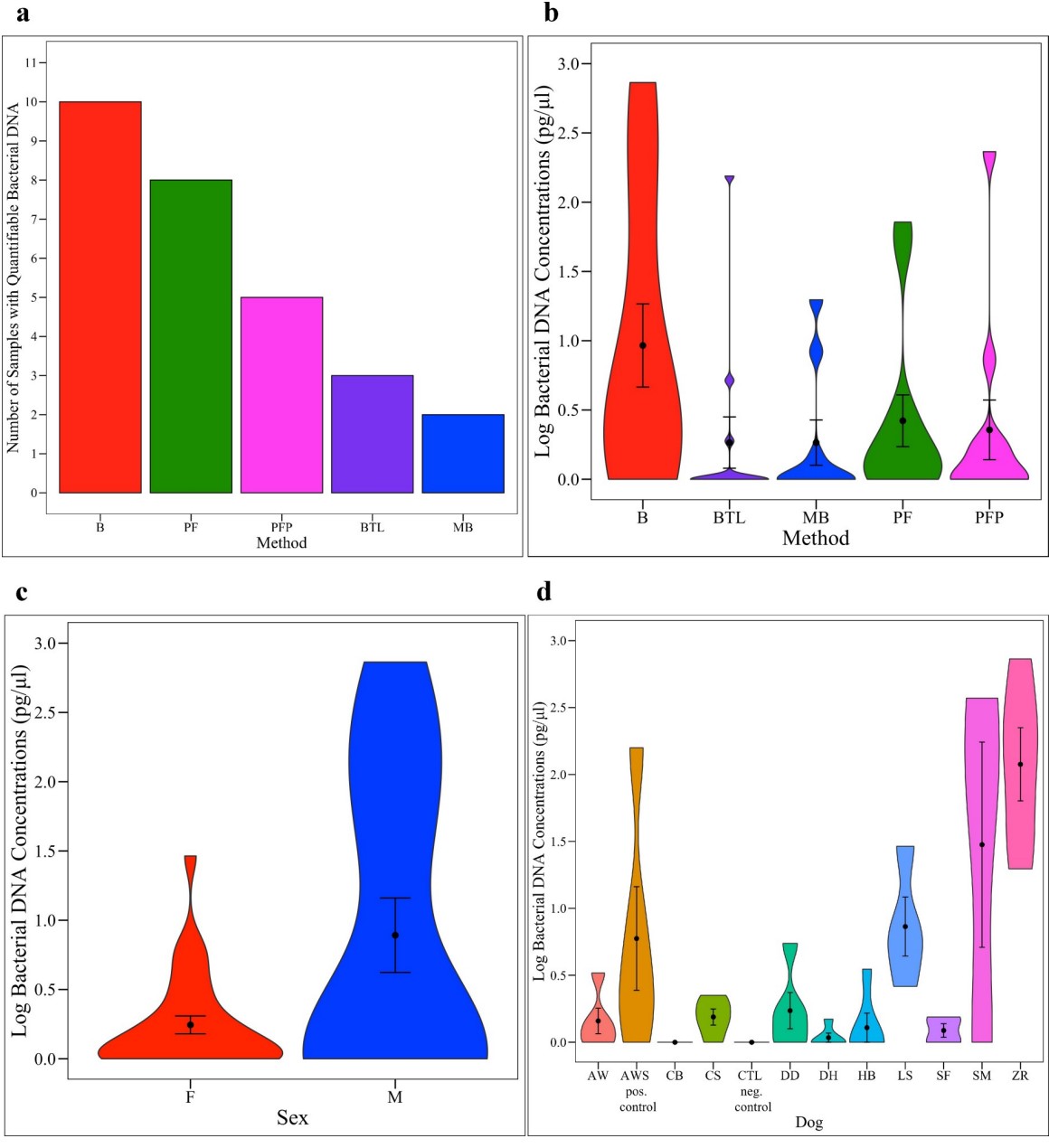

**Fig 2. Bacterial DNA concentrations.** (**a**) Number of samples with measurable bacterial DNA by extraction method. Bacterial DNA concentrations, calculated via qPCR, by (**b**) extraction method, (**c**) sex and (**d**) dog. Bacterial DNA concentrations differed significantly by extraction method (Kruskal-Wallis, *p* = 0.044) and by dog (Kruskal-Wallis, *p* = 0.0005); although, no pairwise comparisons were significant. Bacterial DNA concentrations did not differ significantly by sex (Kruskal-Wallis, *p* = 0.333; positive control excluded). B = Bacteremia, BTL = Blood & Tissue with Lysozyme, MB = Magnetic Beads, PF = PowerFecal®, PFP = PowerFecal® Pro, F = Female, M = Male.

SM (3.4% variation) extracted using the B kit. All three samples were included in analyses. Quantifiable bacterial DNA concentrations ranged from 0.28 pg/μl to 729.48 pg/μl. The number of samples with quantifiable bacterial DNA varied by extraction method (**Fig 2A**). Additionally, bacterial DNA concentrations differed significantly by method (**Fig 2B**, Kruskal-Wallis, *p* = 0.044) and by dog (**Fig 2D**, Kruskal-Wallis, *p* = 0.0005); although, no pairwise

comparisons by method or dog were significant (S4 Table). B yielded the highest bacterial DNA concentrations and extracted quantifiable bacterial DNA from the greatest number of urine samples (Fig 2A). Bacterial concentrations did not differ significantly by sex (Fig 2C, Kruskal-Wallis Rank Sum Test, $p = 0.333$). Bacterial and total DNA concentrations were significantly correlated (S1A Fig, R = 0.42, $p = < 0.001$), and 15 samples were identified as having greater bacterial DNA concentrations than total DNA concentrations (S2 Table).

## Bacterial diversity by extraction method, dog, and sex

Based on decontam (R package, v.1.10.0), we bioinformatically removed a total of 28 putative contaminant ASVs (S5 Table). We also removed all sequences identified as chloroplasts, mitochondria, eukaryotes, and archaea. Additionally, four samples with fewer than 300 reads were excluded from 16S rRNA bacterial community analyses (S2 Table). Three of these four samples came from dog CB, who had one of the lowest urine bacterial concentrations. Three of the 5 negative controls also had fewer than 300 reads while the remaining 2 negative controls had 7,899 (MB) and 6,936 (BTL) reads. Over 99% of the reads in the MB negative control were aligned to the same ASV (*M. plutonius*) that was spiked into our positive controls. No total or bacterial DNA was recovered from this sample after extraction, so we attribute the presence of the positive control taxa in this sample to potential cross contamination during DNA plating, library preparation, or sequencing. The BTL negative control with 6,936 reads contained a total of 122 ASVs, but none of these ASVs were identified in any other sample. The remaining samples ranged from 316 to 42,090 reads (average = 14,721). A total of 51 samples were retained for analysis. The number of 16S rRNA reads per sample differed significantly by dog (Kruskal-Wallis, $p = < 0.001$), but not by sex (Kruskal-Wallis, $p = 0.937$) or extraction method (Kruskal-Wallis, $p = 0.378$); although, B yielded the greatest number of 16S reads per sample (S2 Fig). There was also a significant correlation between the number of 16S reads and bacterial DNA concentrations (S1B Fig, R = 0.28, $p = 0.047$).

Bacterial diversity was compared across samples by extraction method, dog, and sex (Shannon Index: Fig 3; Observed Features and Simpson Index: S3 Fig). The positive control, which was urine from female dog AW spiked with *M. plutonius*, was removed from all analyses by sex to prevent bias. All three measures of diversity revealed the same patterns. Bacterial

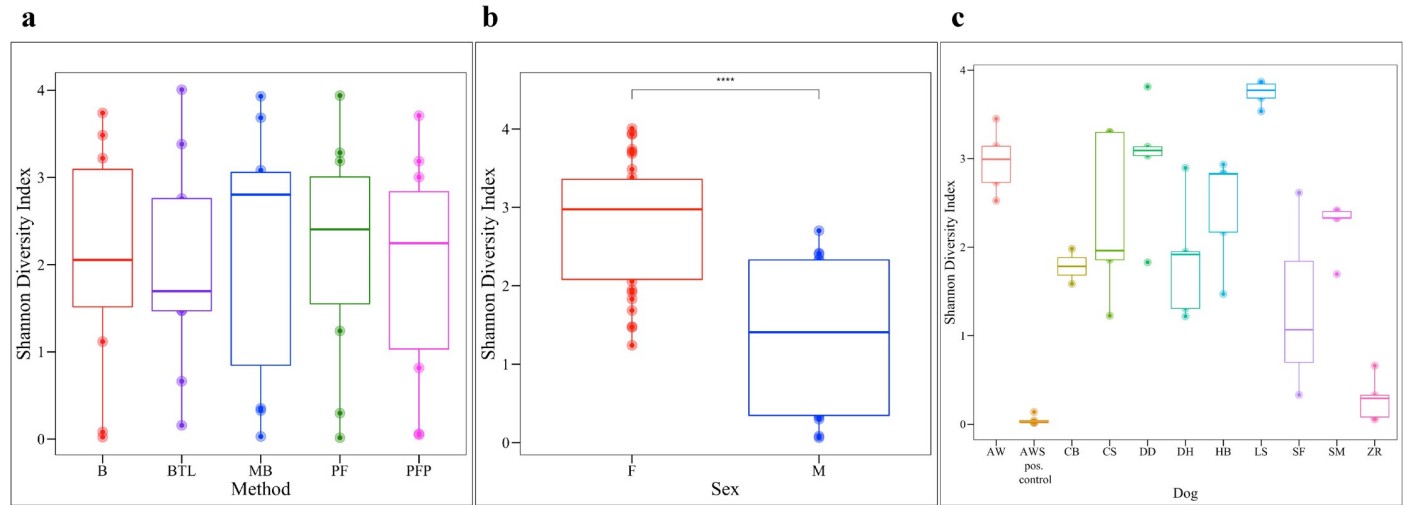

**Fig 3. Bacterial diversity.** The Shannon diversity metric was used to compare bacterial diversity by (**a**) extraction method, (**b**) sex, (**c**) and dog. Bacterial diversity did not differ significantly by kit (Kruskal-Wallis, $p = 0.951$) but did differ significantly by dog (Kruskal-Wallis, $p = < 0.001$). For all statistically significant pairwise comparisons by dog, see S6 Table. Females exhibited higher bacterial diversity than males (Kruskal-Wallis, $p = < 0.001$). B = Bacteremia, BTL = Blood & Tissue with Lysozyme, MB = Magnetic Beads, PF = PowerFecal®, PFP = PowerFecal® Pro, F = Female, M = Male.

diversity did not differ by extraction method (Kruskal-Wallis: Shannon, $p = 0.951$; Observed Features, $p = 0.751$, Simpson, $p = 0.872$; **Fig 3A** and **S3A and S3D Fig**) but did differ significantly by sex (Kruskal-Wallis: Shannon, $p = < 0.001$; Observed Features, $p = 0.005$, Simpson, $p = < 0.001$; **Fig 3B** and **S3B and S3E Fig**) and by dog (Kruskal-Wallis: Shannon, $p = < 0.001$; Observed Features, $p = < 0.001$, Simpson, $p = < 0.001$; **Fig 3C** and **S3C and S3F Fig** and **S6 Table**). Females had significantly higher bacterial diversity than males across all diversity metrics. By dog, LS had significantly higher bacterial diversity than 9 other dogs while ZR had significantly lower bacterial diversity than 9 other dogs. For a full list of significant pairwise comparisons by dog, see **S6 Table**.

## Bacterial composition by extraction method, dog, and sex

Bray Curtis (**Fig 4**) and Unweighted and Weighted UniFrac metrics (**S4 Fig**) were used to compare bacterial composition (beta-diversity) across groups. No significant differences were observed by extraction method (PERMANOVA: Bray Curtis, $p = 1.000$; Unweighted UniFrac, $p = 0.539$; Weighted UniFrac, $p = 0.743$, **Fig 4A** and **S4A and S4D Fig**). However, bacterial composition did differ significantly by sex (PERMANOVA: Bray Curtis, $p = < 0.001$, Unweighted UniFrac, $p = 0.003$; Weighted UniFrac, $p = 0.02$ **Fig 4B** and **S4B and S4E Fig**) and by dog (PERMANOVA: Bray Curtis, $p = < 0.001$, Unweighted UniFrac, $p = < 0.001$; Weighted UniFrac, $p = < 0.001$; **Fig 4C** and **S4C and S4F Fig** and **S7 Table**).

## Bacterial taxonomic differences by extraction method, dog, and sex

In total, there were 23 phyla, 318 genera, and 206 amplicon sequence variants (ASVs, roughly equivalent to species) observed across all canine urine samples. Collectively, the three most abundant phyla across all samples were Proteobacteria, Firmicutes, and Bacteroidetes (**Fig 5**). We also noted that the positive control sample, AWS, which was urine from dog AW spiked with *M. plutonius*, had significantly lower bacterial diversity (Shannon, Wilcoxon Rank Sum test, $p = < 0.001$) as compared to AW. This indicates that in urine dominated by a specific microbe (e.g. during a urinary tract infection), amplicon sequencing precludes the ability to detect other microbes present in the bacterial community. At the phyla level, there were no

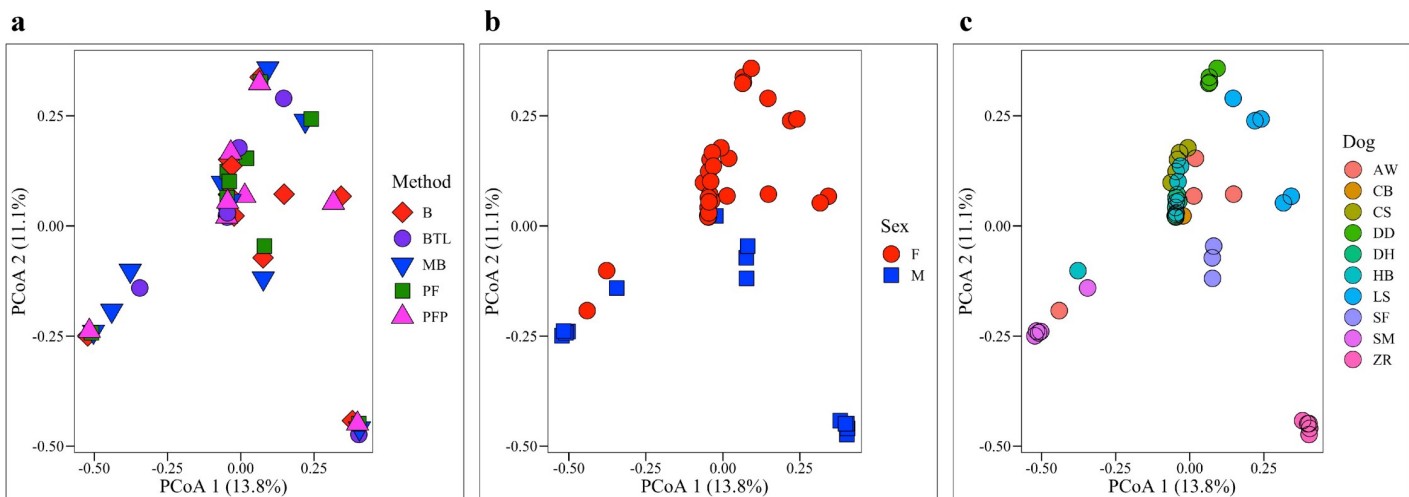

**Fig 4. Bacterial composition.** Bray Curtis dissimilarity matrices were used to compare bacterial composition (beta-diversity) by (**a**) extraction method, (**b**) sex, (**c**) and dog. Bacterial composition did not differ significantly by method (PERMANOVA, $p = 1.000$) but did differ significantly by sex (PERMANOVA, $p = < 0.001$) and dog (PERMANOVA, $p = < 0.001$).

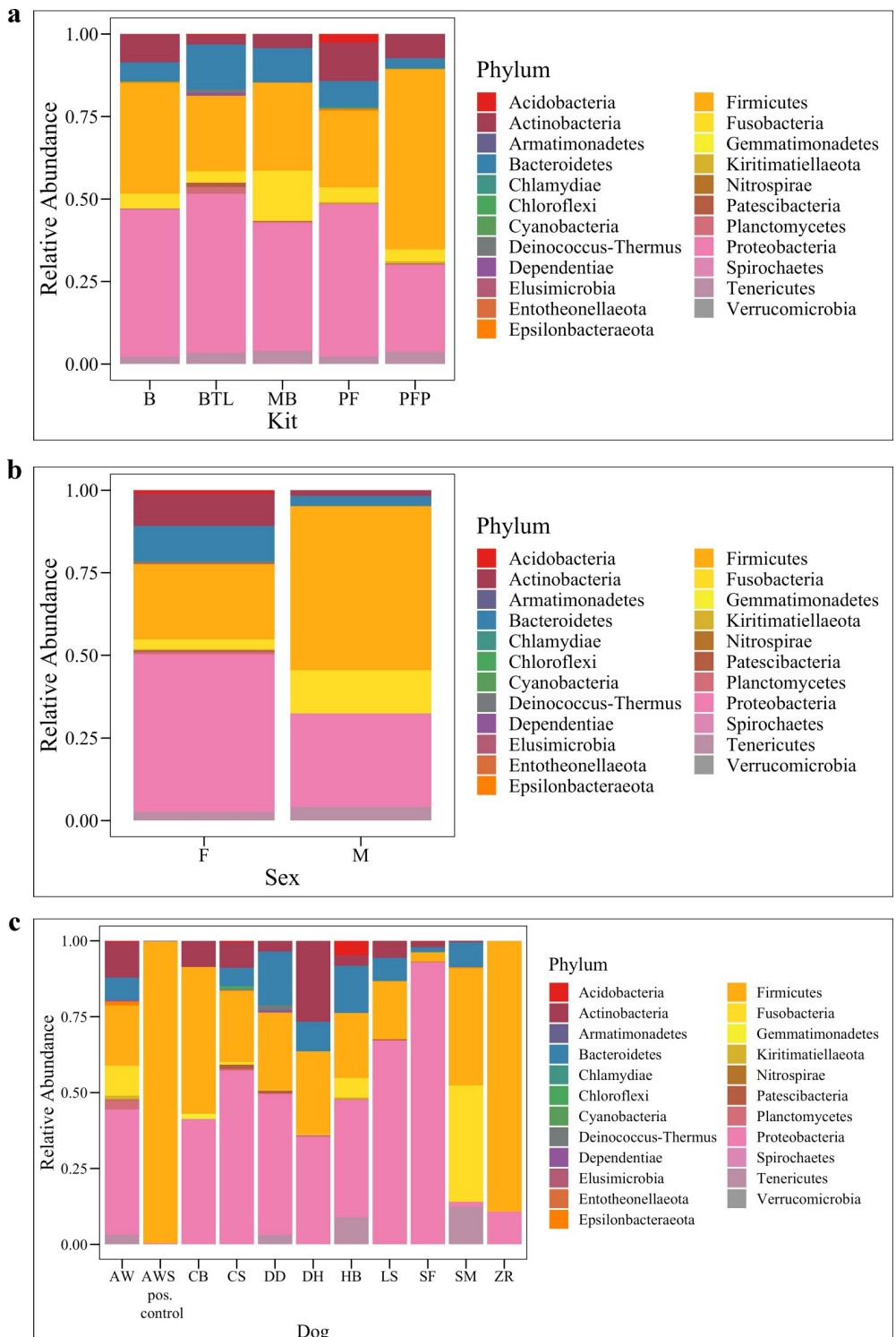

**Fig 5. Bacterial relative abundances by phyla.** Bacterial taxa bar plots by Phyla for (**a**) extraction method, (**b**) sex and (**c**) dog. B = Bacteremia, BTL = Blood Tissue with Lysozyme, MB = Magnetic Beads, PF = PowerFecal®, PFP = PowerFecal® Pro, F = Female, M = Male.

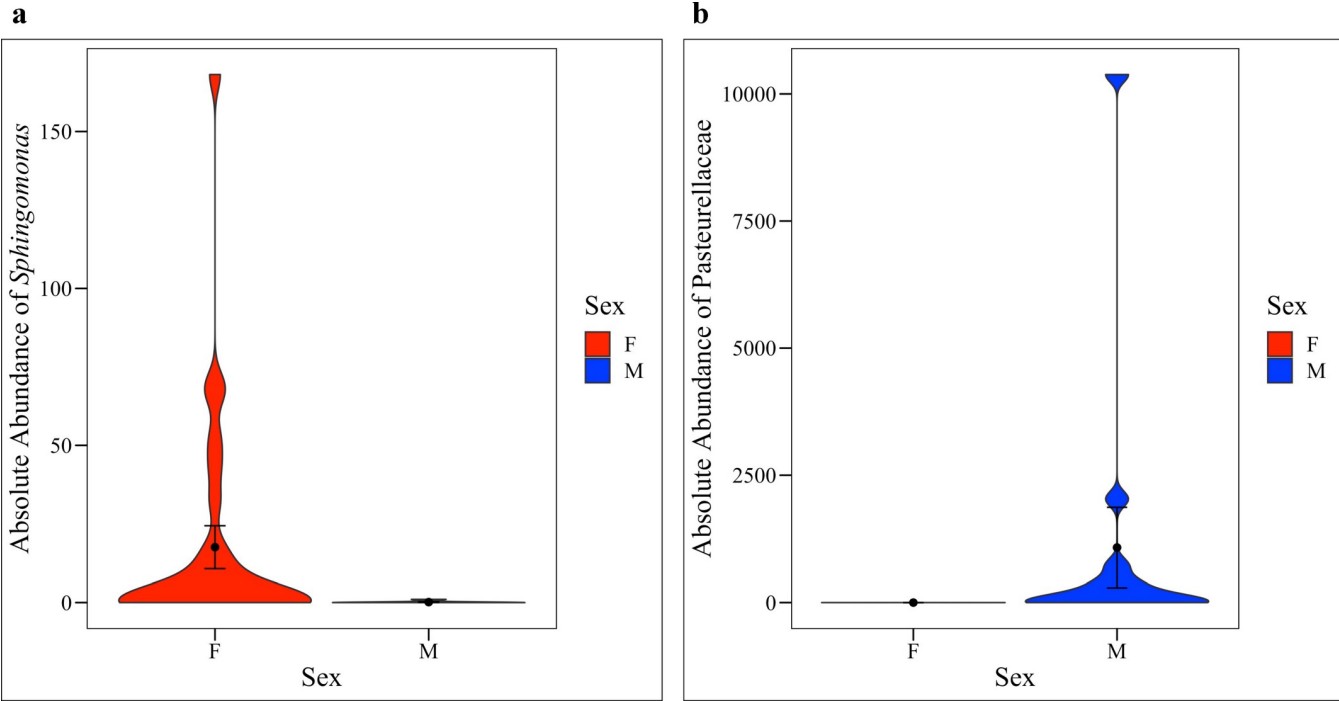

**Fig 6. Differentially abundant taxa by sex.** (**a**) Females had significantly greater absolute cell counts of *Sphingomonas* (Kruskal-Wallis, *p* = 0.0066) while (**b**) males had significantly greater absolute cell counts of *Pasteurellaceae* bacterium canine oral taxon 272 (Kruskal-Wallis, *p* = < 0.001). Only samples with > 300 reads and at least 2 qPCR replicates are included in this figure.

significant differences in taxa abundances by extraction method (**Fig 5A**, Kruskal-Wallis, *p* = 0.81) but there were by sex (**Fig 5B**, Kruskal-Wallis, *p* = 0.02), and dog (**Fig 5C**, Kruskal-Wallis, *p* = < 0.01). By sex, females had a significantly higher abundance of Actinobacteria as compared to males (**Fig 5B**, ANCOM, *W* = 20). At the L7 (roughly species) level, no taxa were found to be differentially abundant by extraction method. However, several taxa were identified as differentially abundant by sex and by dog. By sex, the relative abundance of *Sphingomonas* was significantly increased in females compared to males (**S5A Fig**, ANCOM *W* = 579) while *Pasteurellaceae* bacterium canine oral taxon 272 was significantly more abundant in males (**S5B Fig**, ANCOM *W* = 596). Thirty-two taxa (L7 level) were differentially abundant by dog (**S8 Table**, ANCOM).

Per Nadkarni et al. (2002), we then converted the relative abundances into estimated absolute cell counts using bacterial DNA concentrations derived from qPCR and using the weight of DNA in one *E. coli* cell (4.96 fg) as a standard [70]. This conversion calculation assumes that there is one copy of the 16S rRNA gene in each bacteria and that the molecular weight of DNA in bacterial cells of all taxa is 4.96 fg per cell. In fact, 16S copy number and molecular weight varies widely between taxa; however, Nadkarni et al. found these assumptions to be reasonable when comparing cultured cell counts to cell counts based on qPCR [70]. Total cell counts ranged from approximately 10–147,072 cells per sample. We then multiplied total cell counts by the relative abundances of *Sphingomonas* or *Pasteurellaceae* bacterium canine oral taxon 272 to get estimated absolute cell counts of these taxa. The absolute cell counts of *Sphingomonas* were significantly increased in females (**Fig 6A**, Kruskal-Wallis, *p* = 0.0066) while the absolute cell counts of *Pasteurellaceae* bacterium canine oral taxon 272 were significantly increased in males (**Fig 6B**, Kruskal-Wallis, *p* = < 0.001). (Note: only samples with > 300 reads and at least 2 qPCR replicates were included in this analysis).

## Discussion

We compared total and bacterial DNA concentrations as well as 16S rRNA bacterial community sequencing data from the urine of 10 healthy dogs extracted using 5 different DNA isolation methods. Each method employed various mechanical, chemical, and / or thermal lysing techniques. Sex and dog, but not extraction method, significantly affected DNA concentrations and bacterial diversity and composition. The B kit was determined to be one of the most effective methods for urine bacterial DNA extraction.

### DNA concentrations and 16S reads

The B kit extracted the greatest total and bacterial DNA concentrations from the canine urine samples (**Figs 1B and 2B**). Moreover, B extracted quantifiable *bacterial* DNA from the greatest number of samples (10 urine samples, including the positive control) while BTL extracted quantifiable *total* DNA from the greatest number of samples (9 urine samples, including the positive control) (**Figs 1A and 2A**). Additionally, males contained significantly greater total but not bacterial DNA as compared to females (**Figs 1C and 2C**). Total and bacterial DNA concentrations and number of 16S rRNA reads also varied significantly by dog (**Figs 1D and 2D** and **S2C Fig** and **S3 & S4 Tables**). B also produced the greatest number of 16S rRNA reads as compared to other extraction methods (Kruskal-Wallis, $p = 0.378$); however, as 16S rRNA sequencing data are compositional, we do not weigh this finding heavily [81] (**S2A Fig**). Taken together, B demonstrated efficacy over other methods in extracting DNA from dog urine, while biological factors such as sex and dog had strong effects on DNA concentrations.

In 15 samples, bacterial DNA concentrations were greater than total DNA concentrations, and in several of these cases, no DNA was detected by Qubit while bacterial DNA was amplified via qPCR. This could be due to relatively high bacterial loads in these dogs, and the increased sensitivity of qPCR (bacterial DNA concentrations) as compared to Qubit (total DNA concentrations) [82]. In 9 other samples, 5 of which were extracted by BTL (**S2 Table**), there was quantifiable total DNA but no quantifiable bacterial DNA detected, suggesting that these samples may have contained more host than bacterial DNA, or that BTL was more effective in extracting host as compared to bacterial DNA. Despite the lack of quantifiable bacterial DNA in these samples, we obtained 16S rRNA sequencing reads from all 9 samples; although, 3 of these samples were excluded from 16S rRNA analysis for having fewer than 300 reads. This indicates that 16S rRNA sequencing may be more sensitive to bacterial DNA than qPCR [83]; however, the bacterial taxa present in these samples should be reviewed carefully for potential contamination as they could contain low reads and skewed relative abundances. Alternately, the different primer sets used in qPCR and 16S rRNA sequencing, although both considered "universal bacterial primers," may contribute to the differences we observed in bacterial DNA detection between qPCR and 16S rRNA sequencing.

### Bacterial diversity and composition

Bacterial diversity and composition differed significantly by sex and dog but not extraction method (**Figs 3 and 4**), indicating that individual differences in the urine microbiota overwhelmed potential differences introduced by extraction. A total of four samples were excluded from bacterial community analysis for having fewer than 300 reads. All canine urine samples extracted via B and PF (excluding negative controls) contained greater than 300 reads and were retained for analysis. These results again highlight B as a viable extraction method for urine microbiota, as it did not obviously skew bacterial communities while also generating reasonable 16S rRNA yields. We further observed that urine microbiota were highly variable between individuals, a finding that has been reported previously in studies on human and

canine urine [9, 10, 13, 25]. A few dogs stood out in terms of bacterial composition and diversity. ZR, for example, had significantly lower bacterial diversity (Shannon) than most other dogs while LS had significantly higher diversity than most dogs (**S6 Table**). Both ZR and LS also had significantly different bacterial composition (Bray-Curtis) than most other dogs (**S7 Table**).

We also observed that the samples from females had significantly higher bacterial diversity than the samples from males despite males having higher total (**Fig 1C**) and bacterial DNA concentrations (**Fig 2C**). This suggests that males may be shedding more host cells into urine than females. The increased urine bacterial diversity in the samples from females could be due to differing anatomy between sexes, differing hormone profiles, or differing urination habits. In humans, urine microbial diversity results vary by study. In a 2013 study based on free-catch urine, increased diversity was reported in healthy females as compared to males [32], while in a study from 2020 that compared both free-catch and catheterized urine, no differences in microbial diversity were reported between males and females [12]. The literature on canine urine microbiota is limited. There are only two studies, to our knowledge, on the healthy canine urine microbiota. Both employed cystocentesis for sampling, and microbial diversity did not differ by sex [56, 57]. Hormones have also been linked to changes in the fecal microbiome of women and could feasibly be altering the urine microbiota [84]. In addition, urinary behavior differs between male and female dogs with males urinating more frequently than females [85]. It is feasible that urine volume and retention time in the bladder could alter urine composition and the urine microbial community.

## Taxonomic differences

In this study, the three most abundant phyla across all samples were Proteobacteria, Firmicutes, and Bacteroidetes. Other studies on urine microbiota in humans and in dogs report similar findings [10, 32, 33, 86, 87]. There were no differentially abundant taxa at the phyla level by extraction method or dog. By extraction method, Proteobacteria was the most abundant phyla in samples extracted by B (44.7%), BTL (48.0%), MB (38.8%), and PF (46.2%) while Firmicutes was most abundant in samples extracted by PFP (54.6%). We also observed a few rare taxa that were only detected by one or two methods: Taxa in the phyla Dependentiae were only observed in samples extracted by BTL while Chlamydiae was only observed in samples extracted by B or BTL. Acidobacteria were only detected in sample extracted by B (0.012%), BTL (0.96%), and PF (2.59%). These differences, while not statistically significant, suggest potential biological differences–or biases–by kit. Extraction method biases in the lysis and detection of gram-positive versus gram-negative bacteria has been reported previously, and continued evaluation of kit biases in urine microbiota studies is warranted [61].

By sex, the relative abundance of Actinobacteria was significantly higher in the samples from females than in the samples from males. Actinobacteria has also been reported in the urine of human females [6, 10, 32, 33, 87], and in the oral [88] and gut [89, 90] microbiota of dogs. At the L7 (roughly species) level, we observed several differentially abundant taxa by sex and by dog, but not by extraction method. Notably, *Pasteurellaceae* bacterium canine oral taxon 272 was significantly increased in the samples from males while *Sphingomonas* was significantly increased in the samples from females. Taxa in the Pasteurellaceae family have been reported as part of the canine oral [88, 91, 92], nasal [93], and gut microbiota [94]. It is possible that this taxon is introduced into canine urine through licking of the prepuce or penis. As such, this taxon could represent a skin contaminant or could be a true inhabitant of canine urine. Similarly, *Sphingomonas* has been reported in the canine vaginal microbiota [56] and could represent a genital contaminant or true inhabitant of urine. In humans, *Lactobacillus*

species are common vaginal microbes, but studies on urine microbiota collected via catheter demonstrate that similar or identical *Lactobacillus* species are also present and culturable from the bladder and are not just contaminants [6, 95, 96].

There were several limitations to the work performed here. First, mid-stream free-catch urine was used for this study. This collection technique is highly relevant as it is non-invasive and commonly employed in canine health assessments; however, it is subject to contamination by urethral, genital, and skin microbiota. Cystocentesis is another method of sampling urine directly from the bladder, and it reduces the potential for genital and skin contaminants. In one previous study on healthy canine urine sampled via cystocentesis, no significant differences in microbiota were observed between male and female dogs [56]. In contrast, our study observed several differences in bacterial composition and diversity by sex. These differences could be attributed to genital (e.g. vaginal) or skin bacteria that are captured in free-catch urine. A second limitation is that we cannot determine if the DNA and sequences detected in urine samples came from live or dead bacteria. In other words, we may be detecting microbes that are not actually contributing to the urogenital environment. Specialized culture or assessments of microbial function (e.g. metabolomics, proteomics, transcriptomics) are necessary to make this distinction. Third, we did not assess spay / neuter status, age, diet, or breed in relation to the urine microbiota as we were not powered to make these comparisons (**S1 Table**). However, these variables could very well be linked to alterations in microbiota and future studies parsing out these relationships will be necessary.

In this study, we selected the four most commonly used DNA extraction methods across 53 published studies on the urine microbiota. We also included the PFP kit as it was designed by the manufacturer to replace the older PF kit, which was identified as one of the top four extraction methods. Notably, new kits optimized for low biomass extractions and even urine DNA isolation (e.g. Norgen) continue to emerge on the market and should be tested in future studies. We also made a few modifications to the manufacturer protocols which could alter bacterial profile results. Specifically, for all extraction methods that involved bead beating (B, PF, PFP), we performed two rounds of bead beating for 60 seconds at 6 m/s on an MP FastPrep 24. This differs from the instrumentation and/or timing listed in the protocols, but we wanted to ensure consistent bead beating across all methods as bead beating time and intensity can have an effect on bacterial community composition [97]. Finally, these samples were analyzed using 16S rRNA sequencing. We did this to ensure that we could obtain valid sequencing data from a relatively small amount of urine (3 ml). Now that we have established an effective method for urine microbiota extraction, we can pursue deeper sequencing (e.g. whole shotgun metagenomic sequencing) to more fully characterize bacterial genomes and potential bacterial function. Other studies on the urine microbiota have used a range of urine volumes from 1 ml [11, 12, 98] to 30 ml [56, 99]. We opted to test smaller volumes of urine as it is not always feasible to obtain 30 ml of urine from a dog, particularly a small dog. Moreover, future multi-omic studies may require aliquoting of a single urine sample for multiple analyses (e.g. urinalyses, culture, microbiome sequencing, metabolomics, protein analysis, etc.), and optimizing DNA extractions with minimal input can facilitate this. Using larger volumes of urine may yield larger DNA concentrations, which could more readily facilitate deeper sequencing of these samples.

## Conclusion

The Bacteremia (B) kit yielded the highest total DNA concentrations, the highest bacterial DNA concentrations, the greatest number of 16S rRNA sequencing reads, and it extracted bacterial DNA from the greatest number of samples. Moreover, bacterial diversity and

composition did not significantly differ by kit indicating that no method, including B, dramatically biased the sequencing results. As such, Bacteremia (B) proved effective as an extraction method for studies of the urine microbiota.

## Supporting information

**S1 Fig. Correlation analyses.** Correlation between (**a**) total and bacterial DNA concentrations and (**b**) the number of 16S rRNA sequencing reads and bacterial DNA concentrations. (TIF)

**S2 Fig. 16S rRNA sequencing reads.** The number of 16S reads per sample was compared by (**a**) extraction method, (**b**) sex, and (**c**) dog. There were no significant differences in the number of reads by method (Kruskal-Wallis, $p = 0.378$) and sex (Kruskal-Wallis, $p = 0.937$) but there was a significant difference in the number of reads by dog (Kruskal-Wallis, $p = <0.00001$). (TIF)

**S3 Fig. Bacterial diversity.** Observed Features and the Simpson index were used to compare microbial diversity by extraction method (**a**, **d**), sex (**b**, **e**), and dog (**c**, **f**). Microbial diversity did not differ by extraction method (Kruskal-Wallis; Observed Features, $p = 0.751$; Simpson Index, $p = 0.872$) but did differ by dog (Kruskal-Wallis; Observed Features, $p = < 0.001$; Simpson Index, $p = < 0.001$). For all statistically significant pairwise comparisons by dog, see S6 Table. Females exhibited significantly higher microbial diversity than males (Kruskal-Wallis: Observed Features, $p = 0.005$; Simpson, $p = < 0.001$). B = Bacteremia, BTL = Blood Tissue with Lysozyme, MB = Magnetic Beads, PF = PowerFecal®, PFP = PowerFecal®Pro, F = Female, M = Male. (TIF)

**S4 Fig. Bacterial composition.** Unweighted (**a**-**c**) and Weighted UniFrac matrices (beta-diversity) (**d**-**f**) comparing microbial composition by (**a**, **d**) extraction method, (**b**, **e**) sex, (**c**, **f**) and dog. Microbial composition did not differ significantly by extraction method (Kruskal-Wallis: Unweighted UniFrac, $p = 0.539$; Weighted UniFrac, $p = 0.743$) but did differ significantly by sex (Kruskal-Wallis: Unweighted UniFrac, $p = 0.003$; Weighted UniFrac, $p = 0.03$) and dog (Kruskal-Wallis: Unweighted UniFrac, $p = < 0.001$; Weighted UniFrac, $p = < 0.001$) (TIF)

**S5 Fig. Differentially abundant taxa by sex.** (**a**) Females had significantly greater relative abundances of *Sphingomonas* (ANCOM, $W = 579$) while (**b**) males had significantly greater relative abundances of Pasteurellaceae bacterium canine oral taxon 272 (ANCOM, $W = 596$). (TIF)

**S1 Table. Dog metadata.** Sex, age, and breed of dogs enrolled in this study. (DOCX)

**S2 Table. Sample concentrations and reads.** Total and bacterial DNA concentrations and number of 16S reads in each sample. (16S read counts are reported both before and after running "decontam," an R package that identifies and removes potential contaminant taxa.) B = Bacteremia, BTL = Blood & Tissue with Lysozyme, MB = Magnetic Beads, PF = PowerFecal®, PFP = PowerFecal® Pro. * = Excluded from qPCR analysis as only one replicate amplified in qPCR. ** = Variation between replicates was greater than 3% but samples were included in analysis. *** Excluded from 16S analysis for having < 300 reads. Blue

highlights = no bacterial DNA detected although total DNA was detected.
(XLSX)

**S3 Table. Total DNA concentration pairwise comparisons by dog.** P-values based on Wilcoxon Rank Sum Tests for total DNA concentrations using 1000 permutations and False Discovery Rate corrections. There were no statistically significant pairwise comparisons.
(DOCX)

**S4 Table. Bacterial DNA concentration pairwise comparisons by dog.** P-values based on Wilcoxon Rank Sum Tests for total DNA concentrations using 1000 permutations and False Discovery Rate corrections. There were no statistically significant pairwise comparisons.
(DOCX)

**S5 Table. ASVs identified as contaminants.** We employed both the frequency and prevalence methods in the R package decontam v.1.10.0 to identify putative contaminant ASVs. In total, 28 ASVs were identified and excluded from our analysis as contaminants.
(DOCX)

**S6 Table. Alpha-diversity pairwise comparisons by dog.** P-values resulting from pairwise Wilcoxon Rank Sum Tests for alpha diversity metrics using 1000 permutations and False Discovery Rate corrections. *$p < 0.05$.
(DOCX)

**S7 Table. Beta-diversity pairwise comparisons by dog.** P-values based on PERMANOVA pairwise comparisons using 1000 permutations False Discovery Rate corrections. *$p < 0.05$.
(DOCX)

**S8 Table. Differentially abundant taxa by dog.** Thirty-two taxa at the L7 level were differentially abundant by dog.
(DOCX)

## Acknowledgments

We are grateful to Dr. Sushmitha Durgam for the shared used of her qPCR instrument and to the many dogs and dog owners who participated in this study. We also acknowledge the Ohio Supercomputer Center (Columbus, Ohio, established 1987) for the high performance computing resources used in this study.

## Author Contributions

**Conceptualization:** Vanessa L. Hale.

**Data curation:** Ryan Mrofchak, Christopher Madden, Morgan V. Evans, Vanessa L. Hale.

**Formal analysis:** Ryan Mrofchak, Christopher Madden, Morgan V. Evans, Vanessa L. Hale.

**Funding acquisition:** Vanessa L. Hale.

**Investigation:** Ryan Mrofchak, Christopher Madden.

**Methodology:** Ryan Mrofchak, Christopher Madden, Morgan V. Evans.

**Project administration:** Christopher Madden.

**Resources:** Vanessa L. Hale.

**Supervision:** Christopher Madden, Vanessa L. Hale.

**Validation:** Vanessa L. Hale.

**Visualization:** Ryan Mrofchak, Vanessa L. Hale.

**Writing – original draft:** Ryan Mrofchak, Christopher Madden, Vanessa L. Hale.

**Writing – review & editing:** Ryan Mrofchak, Christopher Madden, Morgan V. Evans, Vanessa L. Hale.

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
