## [Decision Letter · Decision Letter 0]

25 Feb 2021

PONE-D-21-01761

Evaluating Extraction Methods to Study Canine Urine Microbiota

PLOS ONE

Dear Dr. Hale,

Thank you for submitting your manuscript to PLOS ONE. After careful consideration, we feel that it has merit but does not fully meet PLOS ONE’s publication criteria as it currently stands. Therefore, we invite you to submit a revised version of the manuscript that addresses the points raised during the review process.

We look forward to receiving your revised manuscript.

Kind regards,

Peter Gyarmati

Academic Editor

PLOS ONE

Journal Requirements:

2. Please amend either the title on the online submission form (via Edit Submission) or the title in the manuscript so that they are identical.

Reviewers' comments:

Reviewer's Responses to Questions

**Comments to the Author**

1. Is the manuscript technically sound, and do the data support the conclusions?

Reviewer #1: Partly

Reviewer #2: Yes

2. Has the statistical analysis been performed appropriately and rigorously? 

Reviewer #1: Yes

Reviewer #2: Yes

3. Have the authors made all data underlying the findings in their manuscript fully available?

Reviewer #1: No

Reviewer #2: Yes

4. Is the manuscript presented in an intelligible fashion and written in standard English?

Reviewer #1: Yes

Reviewer #2: Yes

5. Review Comments to the Author

Reviewer #1: I have some concerns regarding there exclusion of samples based on negative controls. They decide to discard results from two negative controls and state these to be contaminated by dog samples. However, they do not show any data on this.

Reviewer #2: The authors present a study outlining a high throughput sequencing approach for the study of canine urine microbiota. The authors compared 5 commercially available kits for DNA and bacterial DNA extraction. The authors' primary conclusion is that the kit Bacteriemia method outlined in the manuscript is an unbiased and effective method for studies of urine microbiota. The paper is well written even if some sentences could use reworking to make them less cumbersome and specific corrections are provided below. The journal guidelines to authors have been followed for an original article in terms of length, paper structure, ethics, and conflict of interest statement. The objectives were clearly stated and were conducted with adequate study design and methodology. The validity of the results is reliable.

Moreover, a few methodological studies are available so far on low biomass samples microbiota in small animals. Thus, the work gives some interesting starting points that can be useful to other scientists.

However, some concerns arise in the choice of extraction methods being the urine such a difficult matrix. Nowadays, a great number of kits (so-called microbiome kits) are available, that are more suitable for bacterial DNA extraction from low biomass samples ( urine, CSF, skin, blood...). The choice of extraction methods made up for fecal bacterial DNA extraction (Power Fecal and Power Fecal Pro) or total DNA extraction ( Blood and Tissue) may bring a low alpha and beta diversity due to the choice of a not so sensitive extraction method.

This should be discussed in the limitations of the study paragraph.

Some other minor comments below:

Line 22. It should read “a diverse array of microbes is..”

Line 33. It should read “ The Bacteriemia Kit..”

Line 89. “…spay / neuter status, and diet…” These data were not considered in the subsequent analysis, consider to comment on this or to eliminate.

Line 130, line 138 and line 161. I suppose you mean “the first eluate” instead of “the eluent”. It is not clear if you run the first eluate in the column in order to concentrate and maximize DNA yield or you did a second elution step with fresh buffer. The choice of each of the two methods can affect the quantification and thus the results of very low DNA concentration samples. Please clarify.

Line 213. It should read “ Amplicons were sequenced on the Illumina MiSeq platform using V2 chemistry with 2× 250 bp paired-end reads ….”

Line 261. I suppose it should read “..followed by PF, BTL, MB, and PFP” instead of “..followed by PF, BTL, MB, and PF”.

Some parts of the discussion are redundant and make heavy reading (i.e. “This contrasts a previous study on human urine microbiota which reported higher total DNA concentrations in females as compared to males” in Line 403. Consider to comment on this result or concentrate the discussion (i.e. together with line 446).

Consider within the study limitations paragraph also the small sample size analyzed with many variables (10 dogs, including 4 males and 6 females, 8 of them neutered/spayed which can affect the bacterial composition; the age of the dogs (which can be linked to a variation of bladder microbiota from younger to elder age); multiple breeds (Great Pyrenees, one Labrador Retriever, one Golden Retriever, and mixed breed dogs). Consider adding a statement regarding the exclusion of these variables from the subsequent analysis.

6. PLOS authors have the option to publish the peer review history of their article (what does this mean?). If published, this will include your full peer review and any attached files.

Reviewer #1: No

Reviewer #2: No

---

## [Author Response · Author response to Decision Letter 0]

17 May 2021

Dear Dr. Gyarmati and Reviewers,

We are pleased to share with you a revised version of our manuscript, Evaluating Extraction Methods to Study Canine Urine Microbiota. We thank the reviewers for their helpful comments, questions, and suggestions and believe that we have been able to address all of the concerns and substantially improve the manuscript. We address each comment point-by-point below. (Also see the attached response to reviewers for color coded responses.)

Please feel free to contact me if there is any further information you would like regarding this study. 

Sincerely,

Vanessa Hale MAT, DVM, PhD

Co-Director, Infectious Disease Institute Microbial Communities Program

Associate Director, Center of Microbiome Science

Assistant Professor, Veterinary Preventive Medicine, College of Veterinary Medicine

Hale.502@osu.edu

 

2. Please amend either the title on the online submission form (via Edit Submission) or the title in the manuscript so that they are identical.

>Thank you for catching this. We have updated the title in the manuscript to match the online submission form.

}

Reviewers' comments:

Reviewer's Responses to Questions

Comments to the Author

1. Is the manuscript technically sound, and do the data support the conclusions?

Reviewer #1: Partly

Reviewer #2: Yes

>Additional details about negative controls has now been included.

2. Has the statistical analysis been performed appropriately and rigorously? 

Reviewer #1: Yes

Reviewer #2: Yes

3. Have the authors made all data underlying the findings in their manuscript fully available?

Reviewer #1: No

Reviewer #2: Yes

>The BioProject PRJNA689589 with all of the sequencing data is now publicly available. We initially restricted it from public view with the intent to make it publicly available once published. All DNA concentration and qPCR cT data are included in the supplemental materials.

4. Is the manuscript presented in an intelligible fashion and written in standard English?

Reviewer #1: Yes

Reviewer #2: Yes

5. Review Comments to the Author

Reviewer #1: I have some concerns regarding there exclusion of samples based on negative controls. They decide to discard results from two negative controls and state these to be contaminated by dog samples. However, they do not show any data on this.

>Thank you for this comment. In our initial draft of this manuscript, we evaluated the presence and abundance of each taxa in the negative controls, as well as its presence and abundance in true samples AND the DNA concentrations of these true samples in order to identify contaminants. In our revised manuscript, we found and utilized an R package (decontam) that took this same approach, but did so in a more efficient and systematic way, allowing us to detect additional low concentration contaminants. We employed both the “frequency” and “prevalence” methods (at a threshold of 0.5) in decontam to ensure stringent criteria for contaminant identification. In frequency-based identification, the abundance of contaminants is inversely proportional to the total DNA concentrations in the sample (i.e., samples with low DNA concentrations but high abundance of a particular taxon might indicate this taxon is a contaminant). In prevalence-based identification, the prevalence of a taxa in negative controls versus true samples is compared to identify contaminants. Ultimately, we identified and excluded 28 putative contaminants (now listed in Table S4). We then reanalyzed all of our 16S rRNA results. All of our findings remained the same; but we updated all of the p-values and figures as a result of this reanalysis. 

Information on decontam:

• https://microbiomejournal.biomedcentral.com/articles/10.1186/s40168-018-0605-2#citeas

• https://benjjneb.github.io/decontam/vignettes/decontam_intro.html#introduction

Our methods and results have been updated with the information above:

Methods Line 227-230: Prior to 16S rRNA sequencing analysis, singletons (ASVs with only one read in the entire dataset) were excluded. We then used the R package decontam and applied both the frequency and prevalence (at a 0.5 threshold) methods to identify and remove putative contaminants from our dataset (R package, v.1.10.0)

Results Line 304-305: Based on decontam (R package, v.1.10.0), we bioinformatically removed a total of 28 putative contaminant ASVs (S4 Table).

One of our negative controls (from the Magnetic Bead extraction) contained >7000 reads and was 99.5% composed of a single ASV that matched the ASV (honeybee pathogen M. plutonius) we spiked into our positive control samples. There was 0 total or bacterial DNA present in this sample after initial extraction. This lead us to believe that this negative control was likely cross contaminated during the plating / library prep / sequencing step. The other negative control (from the Blood Tissue Lysis extraction kit) with ~ 7000 reads contained 122 ASVs. None of these ASVs were present in any other sample. As both of these negative controls appear to be aberrations and we selected our 300 read cutoff based on careful consideration of the remaining samples and negative controls.

We have now added this information to the text:

Lines 308-315: Three of the 5 negative controls also had fewer than 300 reads while the remaining 2 negative controls had 7,899 (MB) and 6,936 (BTL) reads. Over 99% of the reads in the MB negative control were aligned to the same ASV (M. plutonius) that was spiked into our positive controls. No total or bacterial DNA was recovered from this sample after extraction, so we attribute the presence of the positive control taxa in this sample to potential cross contamination during DNA plating, library preparation, or sequencing. The BTL negative control with 6,936 reads contained a total of 122 ASVs, but none of these ASVs were identified in any other sample.

Reviewer #2: The authors present a study outlining a high throughput sequencing approach for the study of canine urine microbiota. The authors compared 5 commercially available kits for DNA and bacterial DNA extraction. The authors' primary conclusion is that the kit Bacteriemia method outlined in the manuscript is an unbiased and effective method for studies of urine microbiota. The paper is well written even if some sentences could use reworking to make them less cumbersome and specific corrections are provided below. The journal guidelines to authors have been followed for an original article in terms of length, paper structure, ethics, and conflict of interest statement. The objectives were clearly stated and were conducted with adequate study design and methodology. The validity of the results is reliable.

Moreover, a few methodological studies are available so far on low biomass samples microbiota in small animals. Thus, the work gives some interesting starting points that can be useful to other scientists.

However, some concerns arise in the choice of extraction methods being the urine such a difficult matrix. Nowadays, a great number of kits (so-called microbiome kits) are available, that are more suitable for bacterial DNA extraction from low biomass samples ( urine, CSF, skin, blood...). The choice of extraction methods made up for fecal bacterial DNA extraction (Power Fecal and Power Fecal Pro) or total DNA extraction ( Blood and Tissue) may bring a low alpha and beta diversity due to the choice of a not so sensitive extraction method.

This should be discussed in the limitations of the study paragraph.

>Thank you for this point. We have now reflected this in our limitations paragraph. 

Lines 512-516: In this study, we selected the four most commonly used DNA extraction methods across 53 published studies on the urine microbiota. We also included the PowerFecal Pro kit as it was designed to replace the PowerFecal kit, which was one of the top four extraction methods. Notably, new kits optimized for low biomass extractions or even urine DNA isolation continue to emerge on the market and should be rigorously tested in future studies.

Some other minor comments below:

Line 22. It should read “a diverse array of microbes is..”

>This change has been made

Line 33. It should read “ The Bacteriemia Kit..”

>This change has been made

Line 89. “…spay / neuter status, and diet…” These data were not considered in the subsequent analysis, consider to comment on this or to eliminate.

>We have now removed this mention from the methods and note in this as a limitation of our study in the discussion.

Line 508-511: Third, we did not assess the effects of spay / neuter status, age, diet, or breed in this study as we were not powered to make these comparisons (Table S1). Future studies are necessary to determine if or how these variables affect the urine microbiota.

Line 130, line 138 and line 161. I suppose you mean “the first eluate” instead of “the eluent”. It is not clear if you run the first eluate in the column in order to concentrate and maximize DNA yield or you did a second elution step with fresh buffer. The choice of each of the two methods can affect the quantification and thus the results of very low DNA concentration samples. Please clarify.

>Yes, we ran the eluate in the column twice to maximize DNA concentrations. The wording has been changed from “the eluent” to “the first eluate”

Line 213. It should read “Amplicons were sequenced on the Illumina MiSeq platform using V2 chemistry with 2× 250 bp paired-end reads ….”

>We have updated the text with the suggested wording.

Line 261. I suppose it should read “..followed by PF, BTL, MB, and PFP” instead of “..followed by PF, BTL, MB, and PF”.

>This change has been made

Some parts of the discussion are redundant and make heavy reading (i.e. “This contrasts a previous study on human urine microbiota which reported higher total DNA concentrations in females as compared to males” in Line 403. Consider to comment on this result or concentrate the discussion (i.e. together with line 446).

>We have gone back through the entire discussion and edited to reduce redundancy, improve clarity, and simplify the text in several areas including the lines mentioned above.

Consider within the study limitations paragraph also the small sample size analyzed with many variables (10 dogs, including 4 males and 6 females, 8 of them neutered/spayed which can affect the bacterial composition; the age of the dogs (which can be linked to a variation of bladder microbiota from younger to elder age); multiple breeds (Great Pyrenees, one Labrador Retriever, one Golden Retriever, and mixed breed dogs). Consider adding a statement regarding the exclusion of these variables from the subsequent analysis.

>We have now noted within the discussion that we were not powered to evaluate these variables

Line 508-511: Third, we did not assess the spay / neuter status, age, diet, or breed in relation to the urine microbiota as we were not powered to make these comparisons (Table S1). However, these variables could very well be linked to alterations in microbiota and future studies parsing out these relationships will be necessary.

6. PLOS authors have the option to publish the peer review history of their article (what does this mean?). If published, this will include your full peer review and any attached files.

Do you want your identity to be public for this peer review? For information about this choice, including consent withdrawal, please see our Privacy Policy.

Reviewer #1: No

Reviewer #2: No

Evaluating Extraction Methods to Study Canine Urine Microbiota

The authors have submitted a well written manuscript, describing an interesting study. Studies on the urinary microbiota are starting to get published, but so far, they have mostly been limited to humans. Even then, low DNA yields are common in these studies, and thus the protocols involved need to be improved further. I do, however, have a few comments on the manuscript.

Abstract

L. 19: insert may into the sentence – “The urinary microbiota is the collection of microbes present in urine that MAY play a role in host health”, since this still needs to be proven. 

>This change has been made.

Introduction

L. 51: change microbiome to microbiota

>This change has been made

L. 50-52: delete the last part of the sentence “and distinct from contaminants”, since by the nature of your collection method, you will get a lot of contaminants and can therefore not rule this out. 

>This change has been made

L. 63: A recent paper on this topic has been published and should be referenced – Ammitzbøll et al., 2021. Pre- and postmenopausal women have different core urinary microbiota. Sci rep. 11(1):2212. 

>Thank you for pointing this out. This reference has now been added within the Introduction.

Line 62: Despite this, several recent studies on urine have identified clear shifts in the microbial community associated with age (10,26–31)…

L. 78-79: “which are a valuable and translational model for human urinary tract diseases (54)” - This is a repeat of l. 69-70. Just delete it.

>This change has been made.

Methods

L. 96: “All urine samples were stored at -80 °C within 6 hours of urination”. How were they stored before this? 6 hours of room temperature storage can be critical for bacterial samples. This impact may be amplified due to the low biomass of bacteria in the urine samples. 

>All samples collected in this study were stored on ice from collection until before being placed in a -80 °C freezer. This information has been added to the manuscript.

Line 95 – 97: All urine samples were stored on ice immediately after collection and transported to the laboratory within 6 hours of urination. Samples were then transferred into a -80 °C freezer where they remained until extraction.

Table 1. The authors describe that the QIAamp PowerFecal pro DNA kit only contains a thermal lysis step, but not mechanical lysis. However, manufacturers protocol for PowerFecal Pro DNA kit starts with adding the fecal sample to a lysis buffer (step 1), and vortexing fecal samples with beads for varying time based on the homogenizer used (step 2). As such, the kit combines chemical and mechanical lysis, but not thermal lysis. Based on the description of the protocol used in the current study, l. 163-172, the authors appear to replace the mechanical lysis included in the kit, with their own version of bead beating. This may have impacted the final DNA yield.

>Thank you for pointing this out. We have updated/corrected Table 1. To clarify, the homogenization step (step 2) of the PFP protocol calls for two rounds of bead beating on a PowerLyzer 24 bead beater. Our modification to this protocol is that we used two rounds of bead beating (6m/s) on an MP FastPrep-24 bead beater. This was the bead beater available to us and we opted to use a consistent bead beating protocol across the 3 methods that called for bead beating (Bacteremia, PowerFecal, PowerFecal Pro) as differences in bead beating can lead to different microbial community profiles: https://www.ncbi.nlm.nih.gov/pmc/articles/PMC4504704/

Table 1. continued: Similarly, the QIAamp PowerFecal DNA kit normally combines thermal lysis (step 4) with mechanical lysis similar to that of QIAamp PowerFecal Pro DNA Kit (step 5), but with another lysis buffer. The authors appear to have replaced the original mechanical lysis step with their own mechanical lysis and decided not to use the thermal lysis normally included in the protocol. 

>Thank you for pointing this out. We have updated/corrected Table 1. In our modification of the PowerFecal protocol, we skipped thermal lysing and performed for two rounds of bead beating (6 m/s) on an MP FastPrep-24 bead beater instead of 10 minutes of bead beating in a vortex adaptor (as outlined in the PF protocol). (This protocol modification using an MP FastPrep for bead beating was previously established: https://cebp.aacrjournals.org/content/25/2/407. Note the Qiagen PowerFecal kit was formerly known as the MoBio PowerSoil kit until Qiagen bought out MoBio.)

While optimization is critical, based on the modifications I am not sure whether it can be said that the authors have reliably tested the effects of these two DNA extraction kits for canine urinary microbiota DNA extraction. The authors have stated that they performed these modifications in the table text, but I will recommend mentioning it in the limitations section. Secondly, I believe the authors should correct the table text, to specify that the two kits do normally contain mechanical lysis. If the authors consider the bead beating implemented in the manufacturers protocol to be insufficient, they should include this as an argument for the modifications.

>We have updated the table and now noted this in the limitations section. 

Discussion Lines 516-522: We also made a few modifications to the manufacturer protocols which could alter microbial profile results. Specifically, for all extraction methods that involved bead beating (B, PF, PFP), we performed two rounds of bead beating for 60 seconds at 6 m/s on an MP FastPrep 24. This differs from the instrumentation and/or timing listed in the protocols, but we wanted to ensure consistent bead beating across all methods as bead beating time and intensity can have an effect on microbial community composition.

3 mL of urine is not a lot, especially considering the low DNA yield. While I understand that 30 mL may be a low volume from a small dog, 10 mL should be feasible. 

>We agree that 10 ml is feasible to collect, even in small dogs. In this case we opted for 3ml for a couple reasons: 

• First, if we want to use a single urine sample for multiple analyses (e.g. urinalysis, protein analysis, metabolomics, microbiome analyses, etc.), a 10 ml sample would need to be divided into multiple aliquots. As combined analyses will be critical to future studies on the urine microbiome, we wanted to know if 3ml sample could provide sufficient material for a DNA extraction. 

• Second, this study also provides a starting point for future work using cystocentesis in dogs or cats (or urine microbiome studies in even smaller species – like rodents), in which obtaining 10 ml may be more challenging. This work demonstrates that we can, in fact, use 3ml of urine for DNA extractions; although, we note that this is in dogs, using free catch urine. 

• Finally, there was a precedent in human literature for using <1.5ml of urine for urine microbiome studies. The studies below use as little as 200ul of urine up to 1.5ml of urine for each DNA extraction. Because we were concerned about very low biomass, we opted for 3 ml in our study, but we could potentially test lower urine volumes in the future.

o https://bmcmicrobiol.biomedcentral.com/articles/10.1186/s12866-020-01992-4

o https://pubmed.ncbi.nlm.nih.gov/31992287/

o https://www.nature.com/articles/s41598-019-49823-5

o https://pubmed.ncbi.nlm.nih.gov/33436328/

Do you make replicate DNA extractions or is just one sample from each dog processed? Use of replicates would strengthen your study. 

>One urine sample per dog was divided into five 3 ml aliquots, and only 1 aliquot (and no replicates) were extracted per dog, per kit. We agree the replicates could have been valuable. We vortexed the urine samples prior to and throughout aliquoting to homogenize the aliquots to the best of our ability. In some cases, we were also limited by the amount of urine we obtained and did not have a sufficient amount for aliquots and replicates. Despite the lack of replicates, our results indicate that the 5 aliquots per dog, extracted using 5 different kits, yielded remarkably similar microbial communities, suggesting that aliquots did not differ dramatically by dog.

OTUs vs ASVs. For most of the methods and results sections, the authors refer to sequences using operational taxonomic units (OTUs), whereas from l. 352 they start to refer to the sequences as amplicon sequence variants (ASVs). Did the authors cluster the sample to OTUs, or did they use ASVs? I recommend being consistent with this. 12

>We used ASVs. “Observed OTUs” was the former name of diversity metric (through QIIME) that assessed richness – and we used this metric to analyze ASVs (not OTUs). We have removed all “OTUs” in the text and updated “Observed OTUs” to “Observed Features” and noted that this is equivalent to richness to prevent confusion.

L. 115-117: The naming of samples using the initials of the dogs is a bit confusing, especially given that you also use abbreviations for the five kits. I would recommend using numbers (e.g. 1-10 for the dogs). 

We agree that this was confusing and have clarified the text (see below). We have also removed all instances of full sample names (e.g. SMPFP, AWMB) throughout the text and instead explain in the text the exact sample to which we are referring. For example, instead of saying “All samples exhibited less than 3% variation in cycle threshold values between replicates with three exceptions: AWSMB, SFB, and SMB.” the updated text now reads: ”

Lines 279-282: All samples exhibited less than 3% variation in cycle threshold values between replicates with three exceptions: the spiked positive control extracted using magnetic beads (5.6% variation), and samples from dogs SF (6.1% variation) and SM (3.4% variation) extracted using the Bacterimia kit.”

Methods Lines 113 – 115: Samples were assigned initials unique to each dog (AW, CB, CS, DD, DH, HB, LS, SF, SM, and ZR). Extraction methods were abbreviated as seen in Table 1 (B, BTL, MB, PF, or PFP). 

Results

L. 253: Why do you have 12 samples here? Are there two positive controls? Later e.g. in line 284 you mention 11 samples (including a positive control). In general, it is difficult to follow your number of samples throughout the manuscript. Please be clearer on this. 

>Thanks for pointing this out. In our study, we had 10 dogs, one negative control, and one positive control; thus, a total of 12 samples extracted per kit, but only 11 that contained sample. We have now tried to clarify this throughout the text. Specifically, we removed “out of 12” and “out of 11” throughout the text and instead simply state “X kit extracted DNA from # samples, including the positive control.” (There was no quantifiable total or bacterial DNA in any of the negative controls.)

We also added the following to the results:

Lines 247-248: We extracted DNA from a total 10 dogs using 5 different methods. A single positive and negative control were also extracted using each kit.

L. 299: How did you determine what was considered contaminants? In lines 306-307 you state that two of your negative controls were contaminated by urine samples, but how can you determine this? Maybe reagent contaminants just tend to show up when the DNA concentrations are very low – which could be the case for both negative controls and urine samples. Should this influence your cut-off levels to be around 8200 instead of 300 reads? Please evaluate and comment on this. 

>Thank you for this comment. In our initial draft of this manuscript, we evaluated the presence and abundance of each taxa in the negative controls, as well as its presence and abundance in true samples AND the DNA concentrations of these true samples in order to identify contaminants. In our revised manuscript, we found and utilized an R package (decontam) that took this same approach, but did so in a more efficient and systematic way, allowing us to detect additional low concentration contaminants. We employed both the “frequency” and “prevalence” methods (at a threshold of 0.5) in decontam to ensure stringent criteria for contaminant identification. In frequency-based identification, the abundance of contaminants is inversely proportional to the total DNA concentrations in the sample (i.e., samples with low DNA concentrations but high abundance of a particular taxon might indicate this taxon is a contaminant). In prevalence-based identification, the prevalence of a taxa in negative controls versus true samples is compared to identify contaminants. Ultimately, we identified and excluded 28 putative contaminants (now listed in Table S4). We then reanalyzed all of our 16S rRNA results. All of our findings remained the same; but we updated all of the p values and figures as a result of this reanalysis. 

Information on decontam:

• https://microbiomejournal.biomedcentral.com/articles/10.1186/s40168-018-0605-2#citeas

• https://benjjneb.github.io/decontam/vignettes/decontam_intro.html#introduction

Our methods and results have been updated with the information above:

Methods Line 227-230: Prior to 16S rRNA sequencing analysis, singletons (ASVs with only one read in the entire dataset) were excluded. We then used the R package decontam and applied both the frequency and prevalence (at a 0.5 threshold) methods to identify and remove putative contaminants from our dataset (R package, v.1.10.0)

Results Line 304-305: Based on decontam (R package, v.1.10.0), we bioinformatically removed a total of 28 putative contaminant ASVs (S4 Table).

One of our negative controls (from the Magnetic Bead extraction) contained >7000 reads and was 99.5% composed of a single ASV that matched the ASV (honeybee pathogen M. plutonius) we spiked into our positive control samples. There was 0 total or bacterial DNA present in this sample after initial extraction. This lead us to believe that this negative control was likely cross contaminated during the plating / library prep / sequencing step. The other negative control (from the Blood Tissue Lysis extraction kit) with ~ 7000 reads contained 122 ASVs. None of these ASVs were present in any other sample. As both of these negative controls appear to be aberrations and we selected our 300 read cutoff based on careful consideration of the remaining samples and negative controls.

We have now added this information to the text:

Lines 308-315: Three of the 5 negative controls also had fewer than 300 reads while the remaining 2 negative controls had 7,899 (MB) and 6,936 (BTL) reads. Over 99% of the reads in the MB negative control were aligned to the same ASV (M. plutonius) that was spiked into our positive controls. No total or bacterial DNA was recovered from this sample after extraction, so we attribute the presence of the positive control taxa in this sample to potential cross contamination during DNA plating, library preparation, or sequencing. The BTL negative control with 6,936 reads contained a total of 122 ASVs, but none of these ASVs were identified in any other sample.

L. 360-369 and again l. 463-479: How about genera? Often amplicon sequencing of the 16S rRNA gene focus on the taxonomic levels phylum, family, and genus (L6), and thus it would be interesting to hear if this was affected by extraction method.

>We ran analyses at the phylum, L6, and L7 (roughly species) levels. The phylum level analyses are included in the manuscript (Line 366-369, 469-470) as are the L7 level analyses (Lines370-376, 483-488). L6 (genera) level analyses yielded very similar results to the L7 level analyses. No taxa were differentially abundant by extraction method, 42 taxa were differentially abundant by dog (instead of 32 at the L7 level), and only 2 taxa were differentially abundant by sex – Sphingomonas (increased in females) and Pasteurellaceae bacterium canine oral taxon 272 (increased in males), which is also what we found at the L7 level. As such, we opted to report results and make biological interpretations at the lowest taxonomic level (L7).

Discussion 

L. 419-420: Another example where the description is a bit confusing regarding the number of samples. 

We agree and have gone back through the entire manuscript – including the line numbers noted above - to clarify and simplify all of the areas of the text in which we list sample numbers. 

Supplementary data

S2 table. The font size is a bit small. The table can handle being a bit wider, which would free up space for an increased font size.

>Thank you for pointing this out. The font size has been increased.

---

## [Decision Letter · Decision Letter 1]

9 Jun 2021

PONE-D-21-01761R1

Evaluating Extraction Methods to Study Canine Urine Microbiota

PLOS ONE

Dear Dr. Hale,

Thank you for submitting your manuscript to PLOS ONE. After careful consideration, we feel that it has merit but does not fully meet PLOS ONE’s publication criteria as it currently stands. Therefore, we invite you to submit a revised version of the manuscript that addresses the points raised during the review process.

Please adjust the text according to the suggestions from Reviewer 1.

We look forward to receiving your revised manuscript.

Kind regards,

Peter Gyarmati

Academic Editor

PLOS ONE

Journal Requirements:

Reviewers' comments:

Reviewer's Responses to Questions

**Comments to the Author**

1. If the authors have adequately addressed your comments raised in a previous round of review and you feel that this manuscript is now acceptable for publication, you may indicate that here to bypass the “Comments to the Author” section, enter your conflict of interest statement in the “Confidential to Editor” section, and submit your "Accept" recommendation.

Reviewer #1: All comments have been addressed

Reviewer #2: All comments have been addressed

2. Is the manuscript technically sound, and do the data support the conclusions?

Reviewer #1: Yes

Reviewer #2: Partly

3. Has the statistical analysis been performed appropriately and rigorously? 

Reviewer #1: Yes

Reviewer #2: Yes

4. Have the authors made all data underlying the findings in their manuscript fully available?

Reviewer #1: Yes

Reviewer #2: Yes

5. Is the manuscript presented in an intelligible fashion and written in standard English?

Reviewer #1: Yes

Reviewer #2: Yes

6. Review Comments to the Author

Reviewer #1: Thank you for answering all questions and comments.

A few remarks:

- L. 248: "We extracted DNA from a total 10 dogs" - replace with "We extracted DNA from a total of 10 dogs".

- L. 500-502: Very long sentence and I think something has gone wrong in the sentence structure.

- L. 595-598: Use abbreviations for the kits.

- Several places in the manuscript you switch between using the abbreviation and full name for the kits. Please be consistent.

Reviewer #2: Manuscript Number: PONE-D-21-01761R1

Manuscript Title: Evaluating Extraction Methods to Study Canine Urine Microbiota

General comment: All the suggested comments have been addressed by the authors.

7. PLOS authors have the option to publish the peer review history of their article (what does this mean?). If published, this will include your full peer review and any attached files.

Reviewer #1: No

Reviewer #2: No

---

## [Author Response · Author response to Decision Letter 1]

11 Jun 2021

Dear Dr. Gyarmati and Reviewers,

We are pleased to share with you a second revision of our manuscript, Evaluating Extraction Methods to Study Canine Urine Microbiota. We thank the reviewers for their comments. We have addressed each of the suggestions point-by-point below.

Please feel free to contact me if there is any further information you would like regarding this study. 

Sincerely,

Vanessa Hale

Review Comments to the Author

Reviewer #1: Thank you for answering all questions and comments.

A few remarks:

- L. 248: "We extracted DNA from a total 10 dogs" - replace with "We extracted DNA from a total of 10 dogs".

>Thank you for catching this error. The sentence has been edited.

- L. 500-502: Very long sentence and I think something has gone wrong in the sentence structure.

>This sentence has been edited.

- L. 595-598: Use abbreviations for the kits.

>We have now replaced kit names with abbreviations throughout the manuscript (except when we first introduce the kit names).

- Several places in the manuscript you switch between using the abbreviation and full name for the kits. Please be consistent.

>We have now replaced kit names with abbreviations throughout the manuscript (except when we first introduce the kit names).

Reviewer #2: Manuscript Number: PONE-D-21-01761R1

Manuscript Title: Evaluating Extraction Methods to Study Canine Urine Microbiota

General comment: All the suggested comments have been addressed by the authors.

---

## [Editor Report · Decision Letter 2]

17 Jun 2021

Evaluating Extraction Methods to Study Canine Urine Microbiota

PONE-D-21-01761R2

Dear Dr. Hale,

We’re pleased to inform you that your manuscript has been judged scientifically suitable for publication and will be formally accepted for publication once it meets all outstanding technical requirements.

Kind regards,

Peter Gyarmati

Academic Editor

PLOS ONE
---

## [Editor Report · Acceptance letter]

28 Jun 2021

PONE-D-21-01761R2 

Evaluating Extraction Methods to Study Canine Urine Microbiota 

Dear Dr. Hale:

I'm pleased to inform you that your manuscript has been deemed suitable for publication in PLOS ONE. Congratulations! Your manuscript is now with our production department. 

Kind regards, 

on behalf of

Dr. Peter Gyarmati 

Academic Editor

PLOS ONE